# Liver Fibrosis and Steatosis in Alström Syndrome: A Genetic Model for Metabolic Syndrome

**DOI:** 10.3390/diagnostics11050797

**Published:** 2021-04-28

**Authors:** Silvia Bettini, Giancarlo Bombonato, Francesca Dassie, Francesca Favaretto, Luca Piffer, Paola Bizzotto, Luca Busetto, Liliana Chemello, Marco Senzolo, Carlo Merkel, Paolo Angeli, Roberto Vettor, Gabriella Milan, Pietro Maffei

**Affiliations:** 1Internal Medicine 3, Department of Medicine, DIMED, University of Padua, 35128 Padua, Italy; francesca.favaretto@unipd.it (F.F.); luca.busetto@unipd.it (L.B.); roberto.vettor@unipd.it (R.V.); gabriella.milan@unipd.it (G.M.); pietro.maffei@aopd.veneto.it (P.M.); 2Internal Medicine 5, Department of Medicine, DIMED, University of Padua, 35128 Padua, Italy; giancarlobombonato@unipd.it (G.B.); lucapiffer.tn@gmail.com (L.P.); paolabizz@gmail.com (P.B.); liliana.chemello@unipd.it (L.C.); carlo.merkel@unipd.it (C.M.); pangeli@unipd.it (P.A.); 3Gastroenterology Department of Oncological and Gastroenterological Surgical Sciences, DiSCOG, University of Padua, 35128 Padua, Italy; marcosenzolo@hotmail.com

**Keywords:** Alström syndrome, metabolic syndrome, FIB-4, fibrosis, liver/kidney ratio, NAFLD, shear wave elastography, obesity, diabetes

## Abstract

Alström syndrome (ALMS) is an ultra-rare monogenic disease characterized by insulin resistance, multi-organ fibrosis, obesity, type 2 diabetes mellitus (T2DM), and hypertriglyceridemia with high and early incidence of non-alcoholic fatty liver disease (NAFLD). We evaluated liver fibrosis quantifying liver stiffness (LS) by shear wave elastography (SWE) and steatosis using ultrasound sonographic (US) liver/kidney ratios (L/K) in 18 patients with ALMS and 25 controls, and analyzed the contribution of metabolic and genetic alterations in NAFLD progression. We also genetically characterized patients. LS and L/K values were significantly higher in patients compared with in controls (*p* < 0.001 versus *p* = 0.013). In patients, LS correlated with the Fibrosis-4 Index and age, while L/K was associated with triglyceride levels. LS showed an increasing trend in patients with metabolic comorbidities and displayed a significant correlation with waist circumference, the homeostasis model assessment, and glycated hemoglobin A1c. SWE and US represent promising tools to accurately evaluate early liver fibrosis and steatosis in adults and children with ALMS during follow-up. We described a new pathogenic variant of exon 8 in *ALMS1*. Patients with ALMS displayed enhanced steatosis, an early increased age-dependent LS that is associated with obesity and T2DM but also linked to genetic alterations, suggesting that *ALMS1* could be involved in liver fibrogenesis.

## 1. Introduction

Alström syndrome (ALMS; OMIM #203800) is an ultra-rare, autosomal recessive monogenic disease characterized by a wide spectrum of clinical manifestations involving insulin resistance (IR), multi-organ fibrosis, obesity, type 2 diabetes mellitus (T2DM), hypertriglyceridemia, and hepatic dysfunction [1,2,3]. ALMS is caused by mutations in the *ALMS1* gene [4,5,6,7,8,9,10], which encodes a ubiquitously expressed protein localized to the centrosome and basal bodies of ciliated cells implicated in cytoplasmic microtubular organization, intracellular transport, and cilia assembly or function [11,12,13,14,15].

Liver involvement in ALMS was firstly described with pathological findings of chronic active hepatitis in a child [16]. Later, acute liver failure, hepatocellular carcinoma (HCC), and esophageal varices bleeding were observed in young patients [17,18,19,20]. An evaluation from all over the world reporting hepatic and gastrointestinal findings in 97 patients with ALMS revealed hepatomegaly and splenomegaly, since the age of 8 years, biopsies with hepatic steatosis, bridging fibrosis and cirrhosis and esophageal and gastric varices [21]. Very recently, the first liver transplant in a 19-year-old male patient with ALMS has been described, with a positive follow-up at 2 years [22].

Nonalcoholic fatty liver disease (NAFLD) is characterized by excessive hepatic fat accumulation, is associated with IR and includes non-alcoholic fatty liver (NAFL) and nonalcoholic steatohepatitis (NASH), ranging from fibrosis, cirrhosis to HCC. Fibrosis is the most important prognostic factor in NAFLD, and it is correlated with mortality [23,24]. Liver biopsy is still considered the gold standard in the evaluation of liver fibrosis, even though it is invasive, painful, costly and affected by sampling limitations [25,26]. Therefore, liver biopsy is not the ideal method for repeated assessments of disease progression, and new non-invasive procedures are needed. Several scores and indexes have been validated to detect advanced hepatic fibrosis in patients with NAFLD, such as the alanine aminotransferase (ALT)/aspartate aminotransferase (AST) ratio, the AST-to-platelet ratio index (APRI), and the Fibrosis-4 Index (FIB-4) [27,28,29,30], and innovative biomarkers have been proposed in patients with obesity [31]. Moreover, imaging methods have been developed, such as transient elastography (TE), FibroScan^®^, and shear wave elastography (SWE). SWE is a newer non-invasive method that, like TE, estimates the speed of a shear wave to provide a quantitative measure of liver stiffness (LS) with the advantage of simultaneous anatomic B-mode ultrasound imaging. Different from TE, SWE allows the selection of a liver region devoid of blood vessels or focal lesions for analysis and guarantees a higher accuracy in patients with obesity [32,33,34]. On the other hand, to diagnose and grade hepatic steatosis, the sonographic hepatorenal ratio (or liver/kidney ratio, L/K), is a noninvasive, objective and simple method correlating with histologic samples analysis and proton magnetic resonance spectroscopy measurements [35,36]. 

The aim of our study was, firstly, to quantity liver fibrosis and steatosis in ALMS, using these accurate imaging-derived quantitative estimations (LS and L/K) and correlate them with other clinical and biochemical parameters. Secondly, we analyzed the contribution of metabolic alterations, such as obesity and T2DM, in the progression of NAFLD in this genetic disease, which displays several aspects of the metabolic syndrome and in which fibrosis could play a peculiar and relevant role.

## 2. Materials and Methods

### 2.1. Patients and Control Subjects

Eighteen patients who fulfilled clinical diagnostic criteria for ALMS [37] and carried ALMS1 pathogenetic variants were enrolled at the Internal Medicine 3, Padua University Hospital, in the period between 2017 and 2019. Patients underwent a multi-disciplinary evaluation, and a complete medical history was taken (nutritional aspects, physical activity, smoking and drinking habits, drug and medications, and past and current medical conditions). The diagnosis of T2DM, hypertension, and metabolic syndrome was performed according to recent guidelines [38,39,40]. We recruited 25 healthy volunteers as ultrasound sonographic (US) controls with a negligible daily alcohol consumption. Control subjects had a normal weight (body mass index (BMI): 22.4 ± 4.5 kg/m^2^) and were age-matched with the patients with ALMS (age: 28 ± 8 years). The study was conducted in accordance with the Declaration of Helsinki and approved by the Local Ethics Committee (Prot. n. 2371P); informed written consent was obtained by each patient.

### 2.2. Genetic Analysis

Genomic DNA, obtained by the QIAamp DNA Mini Kit (QIAGEN GmbH, Hilden, Germany) extraction from the peripheral blood of all ALMS patients, was amplified using a standard PCR protocol with the HotStarTaq Master Mix Kit (QIAGEN) using primer sequences firstly for “hot spot” regions of ALMS1 (exons 8, 10, and 16) and, if negative, for all other exons (1–7, 9, 11–15, and 17–23). Amplicons were purified with Illustra ExoProStar (GE Healthcare, Chicago, IL, USA), sequenced using the BigDye Terminator Cycle Sequencing Kit (Thermo Fisher Scientific, Waltham, MA, USA) and analyzed by the 3130xl Genetic Analyzer (Thermo Fisher Scientific). Coding regions and exon–intron boundaries were analyzed; primers sequence and conditions are available on request. Sequences were compared to the GenBank reference sequence NM_015120.4 for ALMS1 using Clustal Omiga, a freely available tool [41]. Pathogenic variants of ALMS1 identified by genomic sequencing were described according to the guidelines of the Human Genome Variation Society (HGVS) reported by den Dunnen et al. [42] and were validated using VariantValidator.org_v0.1.3 [43]. The new variant described in the present study was submitted to the Euro-WABB online database, a locus-specific database (in the Leiden Open Variation Database format) listed by the Human Genome Variation Society in the Locus Specific Mutation Databases LSDBs (www.HGVS.org) [7].

### 2.3. Anthropometric Measurements

All anthropometric measurements were taken with subjects wearing only light clothes without shoes. Height was measured to the nearest 0.01 m using a stadiometer. Body weight was determined to the nearest 0.1 kg using a calibrated balance beam scale. Waist circumference was assessed using a tape measure, and BMI was calculated as weight (kg) divided by the height squared (m^2^). We were not able to collect waist measurements for 3 of the 18 patients.

### 2.4. Biochemical Assessment

For each patient, we measured fasting plasma glucose (FPG), basal insulin, C-peptide, lipid profile ((total cholesterol (TC), high-density-lipoprotein cholesterol (HDL-cholesterol) and low-density-lipoprotein cholesterol (LDL-cholesterol), triglycerides (TG)), platelets, urea, serum creatinine, ALT, AST, and gamma glutamiltrasferase (GGT). All biochemical blood analyses were performed with a standard diagnostic kit according to the WHO First International Reference Standard: fasting glucose (Glucose HK Gen.3, Roche Diagnostic, Indianapolis, IN, USA), insulin, (IMMULITE 2000 Immunoassay, Siemens Healthcare GmbH, Erlangen, Germany), and glycated hemoglobin A1c (Hb1Ac) (HPLC). Platelets were measured by flow cytometry (Sysmex Europe GmbH, Norderstedt, Germany), serum lipids were evaluated by a spectrophotometer (Roche Diagnostic, Indianapolis, IN, USA), and urea, serum creatinine, ALT, AST, and GGT titers were assayed by the enzymatic method with the addition of pyridoxal-5-phosphate in compliance with IFCC reference methods [44]. Glomerular filtration rate (GFR) was estimated by the chronic kidney disease epidemiology collaboration (CKD-EPI) equation. The insulin-resistance index was indirectly estimated using the homeostasis model assessment (HOMA) as following: fasting serum insulin (μU/mL) × fasting plasma glucose (mmol/L))/22.5 [45]. HOMA was not calculated in patients with insulin treatment.

### 2.5. Non-Invasive Fibrosis Markers

The ALT-to-AST ratio (ALT/AST) was calculated by dividing the ALT concentration by the AST concentration (U/L) [27]. The references ranged from, according to our laboratory, 7 to 35 U/L for ALT and 10 to 35 U/L for AST. The APRI was calculated as AST (U/L)/(upper limit of normal)/platelet count (×109/L) × 100 [27,28]. The FIB-4 was calculated using the following equation: (age (years) × AST)/(platelet counts (×10^9^/L) × ALT1/2) [27,29].

### 2.6. Ultrasound Scan

All patients and controls were referred to for an US (Canon Medical System, Aplio i800, probe i8cx1, US band between 4 MHz and 6 MHz) of the abdomen in order to detect the presence and the degree of hepatic steatosis and to assess LS. Each US was performed by the same sonographer of the Internal Medicine 5 at Padua University Hospital. Patients and controls fasted from the midnight of the day scheduled for the scan. SWE measurements were performed on the right lobe of the liver, through intercostal spaces with the patient lying in the supine position with the right arm in the maximal abduction. Measurements were performed at least 1.5 to 2.0 cm beneath the Glisson capsule to avoid reverberation artifacts. The mean value of 5 consecutive measurements was used for statistical analyses. The steatosis diagnosis was based on the L/K consisting in the comparison between the echogenicity intensity measured in the liver (region of interest: approximately 1.2 cm × 1.2 cm) and in the right kidney cortex, sampled at the same depth to reduce the attenuation bias of the two different organs [35,36]. Echo intensity analysis of digitized B-mode images was performed using the software Horos^®^. To determine the presence of steatosis, we considered a cut-off value of 1.6 calculated as mean value (1.2) ± 2 SD (0.2) in our control group. Patients with signs of portal hypertension at the US underwent esophagogastroduodenoscopy, and esophageal varices were classified according the Japan Society for Portal Hypertension [46]. All the measures were performed in accordance with the Italian Association for the Study of the Liver (AISF) [47].

### 2.7. Statistical Analysis

Statistical analyses were performed using the Systat Software SigmaPlot v.13 (Adalta, Arezzo, Italy). All variables were tested by the normality test (Shapiro–Wilk test) and the equal variance test (Brown–Forsythe). Data are presented as the mean value ± SD, when the normality test and equal variance test passed or, if not, as the median value (25th–75th percentile). The Spearman’s correlation coefficient (*r*) and the relative *p*-values were calculated to analyze simple linear correlations between two variables. Comparisons between patients and controls were analyzed by the Mann–Whitney U-test for independent samples and Fisher’s exact test in categorial variables. In all analyses, the *p*-values were two-sided, and a *p*-value lower than 0.05 was considered statistically significant. Skewed data were analyzed by logarithmic transformation.

## 3. Results

### 3.1. Clinical, Biochemical, and Genetic Characterization of Patients with ALMS

Clinical evaluation and biochemical parameters of the 18 patients with ALMS are reported in Table 1.

All patients were genetically characterized, and all ALMS1 variants identified were predicted to cause premature protein truncation; thus, they can be considered true pathogenic variants. We were not able to identify the second ALMS1 pathogenic variant in 3 out of 18 ALMS patients (17%). We described a new pathogenic variant of ALMS1 in exon 8: c.2611_2614delTTCT p.(Phe871Ilefs*10), a deletion of four nucleotides causing a frameshift and predicting a truncated protein of only 871 amino acids (aa) compared with the wild type spanning 4169 aa (Table 2).

The prevalence of overt T2DM in patients with ALMS was 44% (8/18), the prevalence of obesity was 28% (5/18), hypertension was present in 33% (6/18), and the prevalence of metabolic syndrome was 56% (10/18).

Nine patients out of eighteen (50%) showed an increase in transaminases (AST and/or ALT) above the normal range; the three patients (17%) with signs of portal hypertension (splenomegaly or/and esophageal varices) during the US and their esophagogastroduodenoscopy results are described in detail in Table 3. All these three patients presented the criteria for the diagnosis of metabolic syndrome.

### 3.2. Evaluation of LS and Hepatic Steatosis

LS was significantly higher in the patients than in the controls (mean values: 5.3 (range: 4.1–6.5) versus 3.7 (range: 3.3–4.2); *p* < 0.001, Figure 1A), also excluding the three patients with signs of portal hypertension (IDs 1, 2, and 5, described in Table 2 and Table 3 and indicated by white symbols, (mean: 4.8 (range: 4–6.2) versus 3.7 (range: 3.3–4.2); *p* = 0.002). Patients with ALMS displayed L/K values higher than controls (mean: 1.6 (range: 1.2–2) versus 1.3 (range: 1–1.4); *p* = 0.013) (Figure 1B).

Patients with signs of portal hypertension (IDs 1, 2, and 5) described in Table 2 and Table 3 displayed highest LS values and lowest L/K values, as shown Figure 1.

LS was significantly correlated with AST values (r = 0.52, *p* < 0.05) but not with ALT (r = 0.340, *p* = 0.164) or ALT/AST ratios (r = 0.142, *p* = 0.563). Interestingly, LS significantly correlated with FIB-4, the validated non-invasive score for detecting advanced fibrosis (r = 0.590, *p* = 0.012; Figure 2A) and related weakly with APRI, a similar index of fibrosis, (r = 0.45, *p* = 0.067). Surprisingly, we found that LS was correlated with age only in patients with ALMS (r = 0.505, *p* = 0.032; Figure 2B) and not in controls (r = 0.215, *p* = 0.299).

We found a lack of association between steatosis, estimated by L/K, and transaminases values. On the contrary, L/K values were significantly correlated with TG levels (r = 0.504, *p* = 0.032), as shown in Figure 2C.

Lastly, we divided patients according to the presence of steatosis determined by the L/K cut-off value (1.6) calculated in our control group (CTRL; Figure 1) identifying liver steatosis in 10 out of 18 patients with ALMS (56%). LS did not significantly differ between these two subgroups (4.6 (range: 3.1–6.7) versus 5.3 (range: 4.4–6.1) for non-steatosis and steatosis subgroups, respectively; *p* = 0.845), as shown in Figure 3A. Moreover, we did not find any correlation in patients with ALMS, between LS and steatosis estimated by SWE and L/K, respectively (r = –0.065, *p* = 0.792; Figure 3B).

### 3.3. The Role of Comorbidities: Obesity and T2DM

We divided patients with ALMS into subgroups according to the presence of obesity (BMI ≥ 30 kg/m^2^) and T2DM and analyzed the distribution of LS values. LS was not significantly increased in ALMS patients with obesity compared with normal-weight ALMS patients even if we showed an increasing trend (6 (range: 4.8–11.5) versus 4.6 (range: 3.9–6.4), *p* = 0.126; Figure 4A). Likewise, the LS showed an increasing trend in patients with T2DM compared with in patients with normoglycemia, but the difference did not result in statistical significance (6.2 (range: 4.9–9) versus 4.6 (range: 3.6–5.9), *p* = 0.062; Figure 4B).

Nevertheless, the LS showed significant correlations with waist circumference (r = 0.624, *p* = 0.012; n = 15; Figure 5A), HOMA (r = 0.670, *p* = 0.004; n = 15; Figure 5B), and Hb1Ac (r = 0.715, *p* < 0.001; n = 14; Figure 5C) in patients with ALMS.

## 4. Discussion

ALMS displayed severe metabolic phenotypes involving different organ and tissues and can be regarded as a genetic model for obesity, type 2 diabetes, NAFLD and metabolic syndrome.

In particular, the early and quick progression towards cirrhosis in patients with ALMS made the evaluation of hepatic fibrosis an important matter for diagnostic assessment, follow-up, and therapy of this ultra-rare fibrotic disease.

This is the first study that, taking into account the improvements in imaging techniques, adopts SWE to assess the degree of fibrosis and the L/K to quantify the liver steatosis in ALMS. We genetically characterized 18 patients with ALMS describing a new pathogenic variant of exon 8 in ALMS1 and performed in this cohort imaging analysis, biochemical assessment, and expert clinical evaluation. We showed that LS and L/K were significantly higher compared with in the controls, suggesting an increased liver fibrosis and steatosis. In fact, both SWE and L/K demonstrated a good applicability and diagnostic performance [32,33,34,35,36], even if they have never been tested for the diagnosis of liver fibrosis and steatosis, respectively, in patients with ALMS, compared with liver biopsy.

TE was the first tool developed to quantify liver fibrosis by measuring mechanical shear wave propagation through the liver parenchyma [27], recently used in patients with ALMS [48]. However, several studies have demonstrated that SWE is more accurate in the assessment of liver fibrosis compared with TE [27,30,49], both in adults and in children [50].

The high potential of SWE to detect liver fibrosis was enhanced by the correlation between LS and FIB-4 we observed in patients with ALMS [27,29]. Conversely, we did not find any correlation between LS and other serum biomarkers such as ALT/AST ratios and APRI; however, these two well-known scores for fibrosis had a lower accuracy and sensitivity compared to the FIB-4 [30].

Interestingly, we found that LS was correlated with age only in patients with ALMS and not in controls. In the general population, both FIB-4 and LS display an age-dependent increase, respectively, for age greater than 65 [51] and 54 years [52], but this was not the case for our cohort, given all patients were <50 years old. Thus, we could hypothesize that patients with ALMS seem to be older than their age in respect to the fibrotic evolution.

Taken together these results suggest the early appearance of hepatic fibrosis and its rapid progression in patients with ALMS and indicate FIB-4 as a reliable tool to predict fibrosis which is strongly related to LS also in ALMS. The combination of FIB-4 and SWE could improve the prediction of long-term outcomes in patients with suspect advanced fibrosis.

We estimated a 56% of steatosis in patients with ALMS by the L/K, which is considered a useful method to diagnose and grade hepatic steatosis [35,36]. L/K values were significantly correlated with TG levels, confirming TG’s involvement in NAFLD pathogenesis also in ALMS. However, we did not find any relation between LS or L/K and ALT levels, and it could be explained by the normal ALT levels of patients with NAFLD also during disease progression. Furthermore, although elevated aminotransferases should raise suspicion for NASH, normal levels should not be used to exclude NASH [24].

To study any relation between steatosis and fibrosis in ALMS, we divided patients according to the L/K cut-off value (steatosis and non-steatosis subgroups), and we did not find any difference in their LS values. This interesting result suggests that alterations of the *ALMS1* gene could act as an independent determinant of multi-organ fibrosis [1,13]. In other words, the NAFLD pattern reported in ALMS could be characterized by a predominant predisposition to fibrosing steatohepatitis, beyond the presence of fatty acids infiltration in the liver. The progression from steatosis to fibrosis was enhanced when we considered the features of the three case reports with signs of portal hypertension. It is worth noting that all patients presented high levels of LS and none displayed steatosis. We can assume that the initial presence of steatosis was replaced very early by fibrosis, partly by common mechanisms of NAFLD progression and partly by the specific role of dysfunctional ALMS1 protein in liver cells.

According to the multiple hit hypothesis, diet and environmental and genetic factors, together with IR, obesity, and low-grade inflammation, describe the pathogenesis of NAFLD and the risk of progression to inflammation and fibrosis (NASH) or persistence in a stable stage of disease (NAFLD) [53]. Thus, the prevalence of obesity (28%), T2DM (44%), and metabolic syndrome (56%) in our ALMS cohort has been considered as an additional factor linked with the development of NAFLD/NASH besides the genuine effect of the genetic disease per se. LS increased consensually with the presence of metabolic complications, even if the difference did not result significant, probably due to the small sample size of the subgroups. In fact, LS showed a significant correlation with waist circumference, HOMA, and Hb1Ac, supporting the strong association between liver fibrosis and metabolic complications [23,24] also in ALMS. Furthermore, among patients with ALMS and also among our population has demonstrated an increased prevalence of hyperphagia, which represents probably the primary driver for obesity and its complications [3,37]. It is worth noting that the three cases with worse prognosis for liver disease all suffered from T2DM and metabolic syndrome, while two out of three were affected by obesity.

At present, there are no predictive parameters which could be used to evaluate the progression from NAFL to NASH and advanced fibrosis as well as to explain why some patients with ALMS develop serious liver disease while others do not. The pathogenesis of NASH-related fibrosis in the general population is not well-known and even more in ALMS. We previously showed, in fibroblast primary cultures obtained from patients with ALMS, that both an excessive extracellular matrix production and a failure to eliminate myofibroblasts could represent key mechanisms [13].

In Alms1 mutant (foz/foz) mice, it has been demonstrated that bone marrow-derived macrophages (BMMs) contributeS to the hepatic macrophage accumulation [54]. Moreover, the activation of the AP1 transcription factor c-JUN in the pathologic fibroblasts has been described as a possible unified mechanism of an apparently different fibrotic disease [55]. Recently, Geberhiwot et al. showed that patients with ALMS displays insulin resistance at different tissue levels (adipose tissue, liver, and skeletal muscle) compared with BMI-matched controls, providing some evidence, in a new mouse model, that adipose tissue could represent the main driver for metabolic dysregulation in ALMS [56,57]. Thus, several mechanisms could work together, and further studies will be required to link ALMS1 loss of function with metabolic alterations of patients with ALMS.

As a limitation of this study, we have not been able to correlate SWE results with the histologic pattern or the degree of fibrosis with a pathological scoring system, because we did not carry out liver biopsies in patients with ALMS. However, biopsy is not an ideal test to propose and repeat many times during the follow-up, especially for pediatric patients [25,26]; therefore, it could be very useful to find a reliable non-invasive method for assessing hepatic fibrosis [30].

In conclusion, our study showed that SWE examination is a promising diagnostic tool to predict liver fibrosis stage that could further be reinforced by the concomitant evaluation of the FIB-4, both in adult patients with ALMS and particularly in children. Patients with ALMS displayed an early increased LS dependent on age, associated with metabolic impairments, but in addition linked to specific genetic alterations. Patients with ALMS displayed also increased steatosis, which was correlated with TG levels but not with the degree of fibrosis. Thus, the classical NAFLD progression from steatosis to fibrosis does not completely explain the liver disease in ALMS, and we could suggest the involvement of ALMS1 deficiency in the liver disease of patients with ALMS. Thus, better knowledge of ALMS1 protein function in liver fibrogenesis could expand the pathways involved in NAFLD/NASH hepatic disease present also in more common metabolic diseases. Lastly, our work provides valuable tools for further longitudinal studies to monitor patients with ALMS in long-term follow-up and to accurately evaluate the response to new promising treatments against liver fibrosis [58].

## Figures and Tables

**Figure 1 diagnostics-11-00797-f001:**
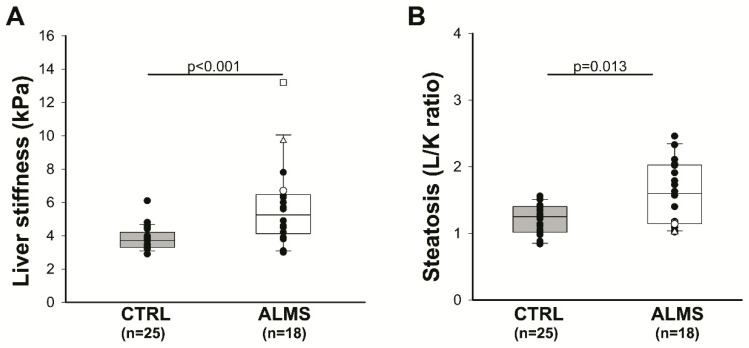
Liver fibrosis and steatosis in patients with ALMS: (**A**) liver stiffness evaluated by shear wave elastography in patients with ALMS and in controls (CTRLs); (**B**) liver steatosis quantified by the L/K in patients with ALMS and in CTRLs. Results are presented as a box plot, with 25th and 75th percentiles and median values. ID 1 (white triangle), ID 2 (white circle), and ID 5 (white square) patients had clinical signs of portal hypertension and are described in detail in Table 3. Statistical analysis was performed using the Mann–Whitney U-test.

**Figure 2 diagnostics-11-00797-f002:**
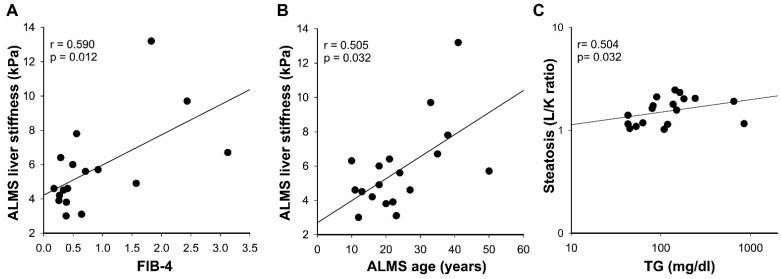
Correlation analysis of the liver stiffness and the sonographic hepatorenal ratio in patients with ALMS. The simple correlations between liver stiffness and FIB-4 (**A**), liver stiffness and age (**B**), and steatosis evaluated by L/K and TG levels (**C**) were performed by Spearman’s correlation in the 18 patients with ALMS. Data are reported on a logarithmic scale in (**C**).

**Figure 3 diagnostics-11-00797-f003:**
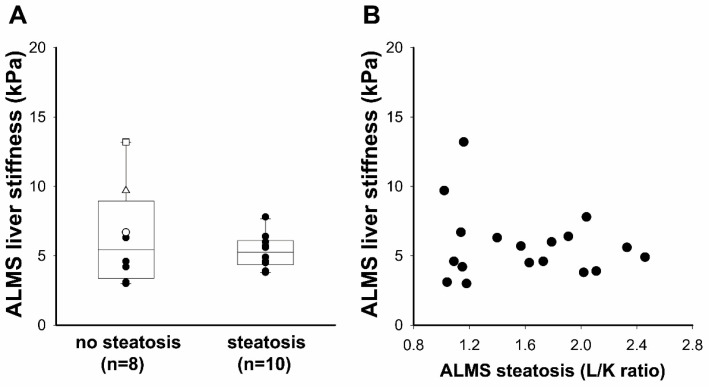
Relationships between liver stiffness and liver steatosis in patients with ALMS: (**A**) liver stiffness evaluated by the shear wave elastography in patients with ALMS divided into two subgroups on the basis of the L/K cut-off value (non-steatosis subgroup with L/K of <1.6 and steatosis subgroup with L/K of >1.6). Results were presented as a box plot, with 25th and 75th percentiles and median values. ID 1 (white triangle), ID 2 (white circle) and ID 5 (white square) patients had clinical signs of portal hypertension and are described in detail in Table 3. Statistical analysis was performed using the Mann–Whitney U-test; (**B**) the simple correlation between liver stiffness evaluated by the shear wave elastography and liver steatosis evaluated by L/K, was performed by Spearman’s correlation in patients with ALMS.

**Figure 4 diagnostics-11-00797-f004:**
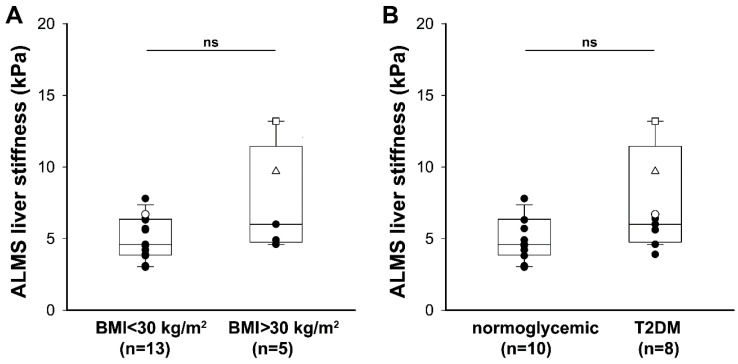
Liver stiffness and metabolic complications in patients with ALMS: (**A**) liver stiffness evaluated by shear wave elastography in patients with ALMS divided into subgroups according to the presence of obesity; (**B**) liver stiffness evaluated by shear wave elastography in patients with ALMS divided into subgroups according to the presence of type 2 diabetes mellitus (T2DM)**.** Data are reported as a box plot with 25th and 75th percentiles and median values. ID 1 (white triangle), ID 2 (white circle), and ID 5 (white square) patients had clinical signs of portal hypertension and are described in detail in Table 3. Statistical analysis was performed using the Mann–Whitney U-test.

**Figure 5 diagnostics-11-00797-f005:**
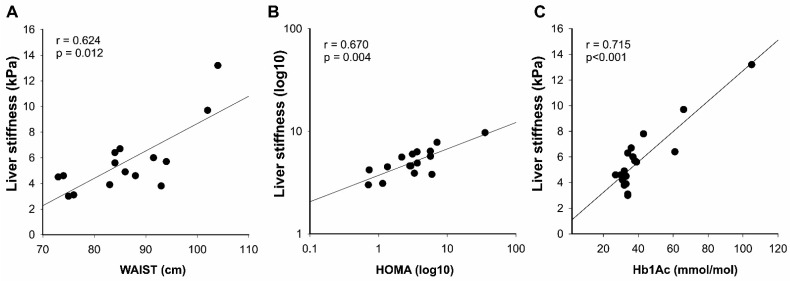
Correlation analysis of the liver stiffness and biomarkers of metabolic complications in patients with ALMS. The simple correlations between liver stiffness evaluated by the shear wave elastography and waist (n = 15) (**A**), and homeostasis model assessment (HOMA; n = 15) (**B**), and glycated hemoglobin 1Ac (Hb1Ac) (n = 14) (**C**) were performed by Spearman’s correlation in the indicated patients with ALMS. Data were transformed into logarithmic values in (**B**).

**Table 1 diagnostics-11-00797-t001:** Anthropometric characteristics and biochemical parameters in 18 patients with Alström syndrome (ALMS). Data are presented as the mean values ± SDs when the normality test (Shapiro–Wilk) and the equal variance test (Brown–Forsythe) were passed or, if not, as the median value (25th–75th percentile).

	Patients with ALMS (*n* = 18)
Sex (M/F)	7/11
Age (y)	24 ± 11
Weight (kg)	63.9 ± 12.7
BMI (kg/m^2^)	27.1 ± 4.3
WC (cm)	86 ± 10
Platelets (×10^9^/L)	215 ± 77
FPG (mg/dL)	4.2 (3.8–6.2)
Insulin (mU/L)	17.7 (10.8–36.7)
C-peptide (μg/L)	3.6 (2.2–6.2)
HOMA	3.5 (2–6.3)
Hb1Ac (mmol/mol)	34 (32–40)
TC (mg/dL)	163 ± 49
HDL (mg/dL)	39 (36–51)
LDL (mg/dL)	100 ± 30
TG (mg/dL)	114 (61–168)
ALT (U/L)	39 (25–73)
AST (U/L)	29 (21–46)
GGT (U/L)	42 (19–61)
Urea (mmol/L)	5.4 (4.6–6.8)
Creatinine (μmol/L)	63 (59–78)
eGFR (mL/min/1.73 m^2^)	117 ± 33
FIB-4	0.49 (0.34–0.92)
APRI	0.36 (0.22–0.71)
ALT/AST	1.23 (1–1.69)

M: male. F: female. BMI: body mass index. WC: waist circumference. FPG: fasting plasma glucose. HOMA: homeostasis model assessment-insulin resistance index. Hb1Ac: glycated hemoglobin A1c. TC: total cholesterol. HDL: high-density-lipoprotein cholesterol. LDL: low-density-lipoprotein cholesterol. TG: triglycerides. ALT: alanine aminotransferase. AST: aspartate aminotransferase. GGT: gamma glutamil transferase. eGFR: estimated glomerular filtration rate. FIB-4: Fibrosis-4 index. APRI: AST-to-platelet ratio index. ALT/AST: ALT-AST ratio.

**Table 2 diagnostics-11-00797-t002:** Description of ALMS1 pathogenic variants in patients with ALMS. Pathogenic variants of ALMS1 identified by genomic sequencing were described as c.DNA variants with respect to the reference sequence NM_015120.4 according to the guidelines indicated by the Human Genome Variation Society (HGVS) described by den Dunnen et al. [42] and were validated using VariantValidator.org_v0.1.3 [43].

**(A)**				
**GENOTYPE**				
**Allele 1**				
**ID**	**Variant**	**Exon**	**Protein**	**Reference**
1	?	?	?	
2	c.7304_7305delAG	8	p.(Glu2435Valfs *7)	Marshall 2015
3 *	c.1046G > A	5	p.(Trp349 *)	Nasser 2018; Weisschuh 2016
4 *	c.1046G > A	5	p.(Trp349 *)	Nasser 2018; Weisschuh 2016
5	c.2164A > T	8	p.(Lys722 *)	Marshall 2015
6 #	c.3019dupA	8	p.(Arg1007Lysfs *15)	Marshall 2015
7 #	c.3019dupA	8	p.(Arg1007Lysfs *15)	Marshall 2015
8	c.1568dupT	8	p.(Ser524Lysfs *13)	Marshall 2015
9 §	c.3425C > G	8	p.(Ser1142 *)	Marshall 2015
10 §	c.3425C > G	8	p.(Ser1142 *)	Marshall 2015
11	c.4937C > A	8	p.(Ser1646 *)	Marshall 2015
12	c.2041C > T	8	p.(Arg681 *)	Dassie 2021
13 +	c.3425C > G	8	p.(Ser1142 *)	Marshall 2015
14 +	c.3425C > G	8	p.(Ser1142 *)	Marshall 2015
15	c.6486_6489delAACT	8	p.(Thr2163Lysfs *4)	Marshall 2015
16	c.10557dupT	16	p.(Pro3520Serfs *5)	Dassie 2021
17	c.3251_3258delCTGACCAG	8	p.(Ala1084Aspfs *3)	Marshall 2015
18	c.3425C > G	8	p.(Ser1142 *)	Marshall 2015
(**B**)				
**GENOTYPE**				
**Allele 2**				
**ID**	**Variant**	**Exon**	**Protein**	**Reference**
1	c.11313_11316delTAGA	16	p.(Asp3771Glufs *20)	Marshall 2015
2	c.10975C > T	16	p.(Arg3659 *)	Marshall 2015
3 *	c.1046G > A	5	p.(Trp349 *)	Nasser 2018; Weisschuh 2016
4 *	c.1046G > A	5	p.(Trp349 *)	Nasser 2018; Weisschuh 2016
5	c.11313_11316delTAGA	16	p.(Asp3771Glufs *20)	Marshall 2015
6 #	c.10830_10831insC	16	p.(Arg3611Glnfs *7)	Marshall 2015
7 #	c.10830_10831insC	16	p.(Arg3611Glnfs *7)	Marshall 2015
8	**c.2611_2614delTTCT**	**8**	**p.(Phe871Ilefs *10)**	**NEW**
9 §	c.3425C > G	8	p.(Ser1142 *)	Marshall 2015
10 §	c.3425C > G	8	p.(Ser1142 *)	Marshall 2015
11	c.11703delA	18	p.(Lys3901Asnfs *8)	Marshall 2015
12	c.5135T > G	8	p.(Leu1712 *)	Marshall 2015
13 +	?	?	?	
14 +	?	?	?	
15	c.6486_6489delAACT	8	p.(Thr2163Lysfs *4)	Marshall 2015
16	c.11580dupT	17	p.(Ile3861Tyrfs *7)	Dassie 2021
17	c.6731delA	8	p.(Asp2244Valfs *24)	Marshall 2015
18	c.9379C > T	10	p.(Gln3127 *)	Marshall 2015

For each variant, the first description in the reference column is reported; the new variant (NEW) described in the present study is indicated in bold and was submitted to the Euro-WABB online database, a locus-specific database (in the Leiden Open Variation Database format) listed by the HGVS in the Locus-Specific Mutation Databases (LSDB) (www.HGVS.org) [7]. We were not able to identify the second ALMS1 pathogenic variant in the patients indicated by the question mark (?). Each patient was indicated by a specific identification number (ID); the symbols *, #, §, and + were used to identify siblings.

**Table 3 diagnostics-11-00797-t003:** Clinical and biochemical evaluation of the three patients with ALMS and signs of portal hypertension.

Patients ID (symbols)	Hb1Ac (mmol/mol)	BMI (Kg/m^2^)	ALT (U/L)	AST (U/L)	GGT (U/L)	Steatosis (L/K)	LS (kPa)	FIB-4	Portal hypertension signs
ID 1 (Δ)	66	33.32	24	25	46	1.02	9.7	2.44	Esophageal varices(F3), splenomegaly
ID 2 (○)	36	26.12	51	55	79	1.14	6.7	3.13	Esophageal varices(F2), splenomegaly
ID 5 (□)	105	32.58	62	86	835	1.16	13.2	1.83	Esophageal varices(F1)

Symbols are used in Figures 1A,B, 3A, and 4A,B. ID: identification number of patients with ALMS reported in Table 2. All three patients had a history of type 2 diabetes mellitus and presented the indicated values of glycated hemoglobin 1Ac (Hb1Ac). BMI: body mass index. ALT: alanine aminotransferase. AST: aspartate aminotransferase. GGT: gamma glutamil transferase. L/K: sonographic hepatorenal ratio. LS: liver stiffness. FIB-4: Fibrosis-4 Index. F1: straight, small-caliber varices; F2: moderately enlarged, beady varices; F3: markedly enlarged, nodular, or tumor-shaped varices (according to the Japan Society for Portal Hypertension [46]).

## Data Availability

Data were extracted from the electronic clinical documentation and patient confidentiality was protected by assigning an anonymous identification code.

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
