# Peer review of "Liver Fibrosis and Steatosis in Alström Syndrome: A Genetic Model for Metabolic Syndrome"

_diagnostics, 2021, doi:10.3390/diagnostics11050797_

Round 1

Reviewer 1 Report

To the authors of the manuscript,

I have found of interest the development of ALS and steatosis/fibrosis considering the metabolic etiology of NAFLD and the implication of ALS development. Thus, I have recommended to ACCEPT AT PRESENT FORM.

Reviewer 2 Report

Bettini et al. provided a very intersting study on liver invlovement and its non-invasive assessment in Alstrӧm syndrome. The study is worth publishing. I have only some minor issues.

OVERALL ISSUE

Liver biopsy remains the gold standard for liver fibrosis diagnosis. I am not sure if Authors can state about the presence of liver fibrosis based on elevated LS in SWE only when they do not compare results with liver biopsy. This issue need to be discussed.

The same reflection regarding liver steatosis assessment in the case of lacking liver biopsy. 1H-MRS was desribed as a relatively accurate and reliable method for diagnosing fatty liver, and was used as the diagnostic criteria for several large-scale studies. Many studies have also shown that 1H-MRS was consistent with liver biopsy, in terms of diagnosing fatty liver. This issue need to be discussed

INTRODUCTION provides a sufficient background.

The AIM of the study was properly designed.

MATERIAL AND METHODS - change the order of paragraphs

RESULTS are clearly presented. I have only some minor issues.

L 146-149: On the contrary, L/K values were significantly correlated with triglycerides (TG) levels (r = 0.504, p = 0.032) (Figure 2C), confirming TG involvement in NAFLD pathogenesis of patients with ALMS.

The latter does not comprise the result.

Did You compare the results of AST and ALT values to the references ranges according to Your laboratory ? If yes, please provide the accurate reference ranges.

DISUSSCION

Regarding my overal issue, I am not sure if You could state (L 215-217) : We showed that the LS and the L/K resulted significantly higher compared with controls, indicating an increased liver fibrosis and steatosis.

Congratulations!
